Evaluation of DNA barcoding reference databases for marine species in the western and central Pacific Ocean

Zhou Yufei yufei.zhou@canberra.edu.au 1
Trujillo-González Alejandro 1
Nicol Simon 1 2
Huerlimann Roger 3
Sarre Stephen D. 1
Gleeson Dianne 1
1 Centre for Conservation Ecology and Genomics, University of Canberra , Canberra , ACT , Australia
2 Oceanic Fisheries Programme, Pacific Community , Noumea , New Caledonia
3 Marine Climate Change Unit, Okinawa Institute of Science and Technology Graduate University , Onna-son , Okinawa , Japan
Waiho Khor
Electronic publication date: 2025 Jul 14
Publication date: 2025
Volume: 13
Electronic Location ID: e19674
Received 2025 Mar 20; Accepted 2025 Jun 9
Copyright: ©2025 Zhou et al.
Copyright year: 2025
Copyright holder: Zhou et al.
License: This is an open access article distributed under the terms of the Creative Commons Attribution License, which permits unrestricted use, distribution, reproduction and adaptation in any medium and for any purpose provided that it is properly attributed. For attribution, the original author(s), title, publication source (PeerJ) and either DOI or URL of the article must be cited.
License URL: https://creativecommons.org/licenses/by/4.0/

Keywords: Reference database, DNA barcoding, COI, Marine macrofauna, Western and Central Pacific Ocean

Funding: The University of Canberra and by the European Union “Pacific-European-Union-Marine-Partnership” Programme (agreement FED/2018/397-941) grant to the Pacific Community European Union and Sweden This work was supported by the University of Canberra and by the European Union “Pacific-European-Union-Marine-Partnership” Programme (agreement FED/2018/397-941) grant to the Pacific Community. This publication was produced with the financial support of the European Union and Sweden. Its contents are the sole responsibility of the authors and do not necessarily reflect the views of the European Union and Sweden. The funders had no role in study design, data collection and analysis, decision to publish, or preparation of the manuscript.

==============================
DNA barcoding is a widely used tool for species identification, with its reliability heavily dependent on reference databases. While the quality of these databases has long been debated, a critical knowledge gap remains in their comprehensive evaluation and comparison at regional scales. Marine metazoan species in the western and central Pacific Ocean (WCPO), a region characterized by high biodiversity and limited sequencing efforts, are an example of this gap. This study developed a systematic workflow to assess mitochondrial cytochrome c oxidase subunit I (COI) barcode coverage and sequence quality in two commonly used reference databases for DNA barcoding: the nucleotide reference database from the National Center for Biotechnology Information (NCBI); and from the Barcode of Life Data System (BOLD). Comparative analyses across marine phyla and WCPO regions identified significant barcode gaps and quality problems, providing insights to guide future barcoding efforts. NCBI exhibited higher barcode coverage, but lower sequence quality compared to BOLD. Quality issues, including over- or under-represented species, short sequences, ambiguous nucleotides, incomplete taxonomic information, conflict records, high intraspecific distances, and low inter-specific distances were identified in both databases, likely resulting from contamination, cryptic species, sequencing errors, or inconsistent taxonomic assignment. The barcode identification number (BIN) system in BOLD demonstrated potential for identifying and addressing problematic records, highlighting the benefits of curated databases. Significant barcode deficiencies and quality issues were observed in the south temperate region of WCPO and phyla such as Porifera, Bryozoa, and Platyhelminthes. Additionally, the COI barcode showed limited species-level resolution for certain taxa, including Scombridae and Lutjanidae. Addressing barcode coverage gaps, improving taxonomic representation, and enhancing sequence quality will be essential for strengthening future barcoding initiatives and advancing biodiversity monitoring and conservation in the WCPO and beyond. This study highlights the need for standardized database curation and sequencing practices to improve the global reliability and applicability of DNA barcoding.

Introduction

Marine ecosystems are rich in species diversity and support multiple ecological and socio-economic services (Hills et al., 2011; Hoegh-Guldberg & Bruno, 2010; Pauly et al., 2002). Recent advancements in DNA barcoding, the use of a standardized short DNA fragment to identify species, have greatly enhanced biodiversity assessments by improving the efficiency and accuracy of taxonomic identification (Bourlat et al., 2013; Bucklin, Steinke & Blanco-Bercial, 2011; Hebert et al., 2003). DNA barcoding, when combined with high-throughput sequencing technologies, enables the efficient investigation of entire biological communities, an approach referred to as DNA metabarcoding (Bourlat et al., 2013; Bucklin, Steinke & Blanco-Bercial, 2011; Stoeckle, Das Mishu & Charlop-Powers, 2020). Both DNA barcoding and metabarcoding are widely used in ecological and biodiversity investigations, including species identification (Baetscher et al., 2023; Bucklin, Steinke & Blanco-Bercial, 2011), dietary analysis (Alberdi et al., 2019; Günther et al., 2021), trophic interaction assessment (Clare, 2014; Leray et al., 2012), and environmental biomonitoring (Bista et al., 2017; Trujillo-González, 2022).

DNA barcoding and metabarcoding require reliable reference databases to ensure the accurate assignment of DNA sequences to specific taxa (Leray et al., 2019; Marques et al., 2021; Ramirez et al., 2020). Global databases like the National Center for Biotechnology Information (NCBI) (Benson, Lipman & Ostell, 1993), the European Nucleotide Archive (ENA) (Kanz, 2004), and the DNA Data Bank of Japan (DDBJ) (Tateno & Gojobori, 1997) are widely used in DNA barcoding studies due to their extensive collections of publicly available sequence records contributed by researchers worldwide. However, the accuracy and reliability of these records have been the subject of debate (Ardura, 2019; Chen, Zobel & Verspoor, 2017; Leray et al., 2019; Marques et al., 2021; Shen, Chen & Murphy, 2013; Turanov & Kartavtsev, 2021). Criticisms of global databases primarily center on whether their curation and validation systems for user-submitted sequences and associated metadata are as robust as specialized curated databases (Ardura, 2019; Chorlton, 2024; Gonçalves & Musen, 2018). For example, some studies have shown that curated reference databases provide more reliable taxonomic identification for DNA barcoding analysis compared to global databases (Gold et al., 2021; Hou et al., 2018; Stoeckle, Das Mishu & Charlop-Powers, 2020). Additionally, redundancies and inconsistencies in global-scale databases can reduce alignment performance, introduce errors, and produce ambiguous results (Chen, Zobel & Verspoor, 2017; Chorlton, 2024).

In comparison, curated databases like the Barcoding of Life Data System (BOLD), which focuses on the mitochondrial cytochrome c oxidase subunit I (COI) barcode (Ratnasingham & Hebert, 2007), and Greengenes (DeSantis et al., 2006) and SILVA (Quast et al., 2012), which focus on ribosomal RNA genes, are generally considered to be more reliable due to their strict quality control protocols and standardized metadata system (Puillandre et al., 2012; Steinke & Hanner, 2011). For instance, the barcode index number (BIN) system (Ratnasingham & Hebert, 2013) is a feature of BOLD that automatically clusters sequences into operational taxonomic units (OTUs) based on genetic similarity, typically corresponding to species-level groupings. This system facilitates species delimitation, highlights potential cases of cryptic diversity, and assists in identifying problematic records, thereby enhancing the reliability of sequence and taxonomy data (Costa et al., 2012; Fontes et al., 2021; Oliveira et al., 2016). However, the lack of barcode records remains a significant challenge in curated databases, potentially reducing taxonomic resolution and increasing the likelihood of misidentification or failed taxonomic assignments (Ardura, 2019; Puillandre et al., 2012; Ramirez et al., 2020). BOLD, in particular, has been reported to exhibit lower public barcode coverage compared to NCBI, partly due to its stricter metadata requirements, voucher specimen standards, and sequence curation protocols, which can limit the immediate availability of sequence submissions (Heller et al., 2018; Hestetun et al., 2020).

The evaluation of databases is normally conducted on BOLD due to its BIN system which facilitates the assessment of record quality (Costa et al., 2012; Fontes et al., 2021; Leite et al., 2020; Ratnasingham & Hebert, 2013). In contrast, the absence of a quality evaluation system in NCBI often limits comparisons between NCBI and BOLD to barcode coverage alone (Mugnai et al., 2021; Ramirez et al., 2020; Stoeckle, Das Mishu & Charlop-Powers, 2020; Weigand et al., 2019). This creates a knowledge gap in the comprehensive evaluation of sequence quantity and quality of both databases. In this study, we developed an approach to simultaneously evaluate the quantity and quality of databases based on key features of DNA barcodes and the barcoding gap concept (Bucklin, Steinke & Blanco-Bercial, 2011; Hou et al., 2018; Shen, Chen & Murphy, 2013).

Here, we constructed an evaluation system to evaluate COI barcode records of marine metazoan species using data from the NCBI and BOLD databases. To do so, we focused on the western and central Pacific Ocean (WCPO) region, one of the world’s most heavily exploited ecosystems with the highest biodiversity worldwide (Costello et al., 2010; Hills et al., 2011; Nicol et al., 2013). DNA barcoding presents an excellent opportunity to improve biodiversity monitoring and assessment in the WCPO (Günther et al., 2021; Trujillo-González, 2022; Yeh et al., 2020). However, there is currently no systematic evaluation of the reliability of DNA barcode reference databases for marine species in the WCPO (Gold et al., 2021; Macheriotou et al., 2019). Of most interest is the mitochondrial COI region because its efficiency and ability to amplify a wide range of metazoans makes it a widely used target for many DNA metabarcoding studies (Leray et al., 2013; Lobo et al., 2013). Our objectives were to: (1) test the feasibility of the evaluation workflow for identifying issues in barcode records; (2) compare the reliability of NCBI and BOLD; and (3) assess barcode coverage and quality across phyla and regions within the WCPO.

Methods

All data retrieval and analyses were conducted in R (version 4.1.2) (R Core Team, 2019) using the RStudio environment (version 2022.2.0.443) (RStudio Team, 2022). Data manipulation utilized base R functions and the dplyr package (version 1.1.4) (Hadley et al., 2023), while data visualizations were performed using the ggplot2 package (version 3.5.1) (Wickham, 2016).

Species records retrieval

Species occurrence records in nine marine phylum (Annelida, Bryozoa, Chordata, Cnidaria, Arthropoda (crustacean subphylum only), Echinodermata, Mollusca, Platyhelminthes, and Porifera) from the WCPO were retrieved from the Ocean Biodiversity Information System (OBIS, https://obis.org) in October 2024 using the robis package (version 2.11.3) (Provoost & Bosch, 2022). Given the ongoing concerns regarding barcode deficiencies in the Southern Hemisphere and tropical regions (Marques et al., 2021; Ramirez et al., 2020), separate species checklists were generated for the tropical (23.5°N to 23.5°S, 140°E to 150°W), north temperate (23.5°N to 50°N, 140°E to 150°W), and south temperate (23.5°S to 50°S, 140°E to 150°W) regions of the WCPO. This regional separation was intended to better detect underrepresented areas and support region-specific recommendations for enhancing barcode coverage.

The checklists were initially filtered using metadata embedded in OBIS records, retaining only records with valid species names, accepted taxonomic status, and marine habitat classifications. Species names were further refined by removing records containing numbers or ambiguous strings, such as “sp.”, “complex.”, “cf.”. Duplicates were removed, and records were grouped by species names. Finally, the validity of species name and their synonyms was verified again using the World Register of Marine Species (WoRMS) through the worrms package (version 0.4.3) (Chamberlain & Vanhoorne, 2023).

Based on overlapping and distinct distributions among regions, species were categorized into seven distribution groups: Tropical only, North temperate only, South temperate only, All regions, North-tropical, South-tropical, and North-south. These categories were designed to facilitate comparisons of barcode coverage across WCPO regions and are not intended to reflect ecological endemism.

Barcode records retrieval

COI barcode records were retrieved from NCBI in October 2024 using the rentrez package (version 1.2.3) (Winter, 2017). Valid species names and synonyms from the WPCO species checklist were used as taxonomic group keywords, while keywords for the COI gene included “COI”, “CO1”, “cytochrome c oxidase subunit I”, “Cox1”, and “COXI”. Taxonomic information for each sequence record was retrieved using accession IDs.

Complete barcode record data, including sequences and specimen information for WPCO species in BOLD, were retrieved in October 2024 via the BOLD Web Services for Public Data Portal (https://v3.boldsystems.org/index.php/resources/api?type=webservices) using phylum names as taxa keyword. Sequences with species names matching the valid species names and/or synonyms from the WPCO species checklist, labeled as “COI-5P” and “COI-3P”, and with a BIN were retained.

Barcode databases evaluation workflow

A standardized process for barcode database evaluation and visualization was developed, comprising six basic components and three barcode taxonomic validations: (1) Barcode coverage assessment (proportion of species in the checklist with/ without barcode records). (2) Calculation of sequence counts per phylum. (3) Sequence length distribution analysis. (4) Detection of ambiguous nucleotide characters (e.g., N, R, Y, S, W, K, M, B, D, H, V), categorized by their positions (only in the at 5′ or 3′ ends, only in the middle region, and in both ends and middle). (5) Taxonomic information completeness analysis. (6) Species representation analysis, categorizing sequences into over-represented (species with > 100 barcode records), under-represented (species with < 3 barcode records), and normally represented (species with 3–100 barcode records) (Costa et al., 2012; Fontes et al., 2021; Leite et al., 2020).

The barcode taxonomic validations include: (1) Identification of conflicting records (sequences assigned to multiple taxonomic groups). (2) Barcoding gap analysis, comparing interspecific and intraspecific distances (Hou et al., 2018; Shen, Chen & Murphy, 2013; Zhang et al., 2017; Zhang & Zhang, 2014). (3) Sequence clustering based on genetic distances to identify problematic sequences and assess barcode resolution.

The script for the barcode evaluation workflow is available at GitHub (https://github.com/xyzzzeno/reference_evaluation).

Evaluation on NCBI and BOLD

Raw COI barcode records for WPCO species from NCBI and BOLD underwent the evaluations after retrieval, without prior cleanup or curation. Barcode coverage for species in the tropical, north temperate, and south temperate WCPO regions was analyzed separately. Each species was classified into four categories based on its barcode coverage: no records in either database (no barcode), records in both BOLD and NCBI (both databases), records only in BOLD (only BOLD), and records only in GenBank (only NCBI). The proportions of barcoded and non-barcoded species were also calculated across distribution categories (species distinct to a region or shared among regions). Two-proportions Z-tests were conducted to compare NCBI and BOLD, and two-way analysis of variance (two-way ANOVA) tests (Sthle & Wold, 1989) were conducted to compare WCPO regions and marine phyla.

The remaining five basic evaluations were performed on NCBI and BOLD records for all WPCO species without dividing them into climate regions or distribution categories. Two-proportions Z-tests were conducted to compare evaluation results between NCBI and BOLD.

Conflict records were initially identified by grouping identical sequences and flagging those sequences assigned to multiple taxonomy information. Barcoding gap analysis was conducted to further assess barcode taxonomic accuracy. Due to the computational intensity of this analysis, subsets of 100 species (at least three species per phylum) were randomly sampled from each database. Subsets were aligned using MAFFT (Katoh, 2002), and neighbor-joining (NJ) trees were constructed using the Jukes-Cantor (JC) model (Jukes & Cantor, 1969) in Geneious software (version 2023.2.1) (https://www.geneious.com). Pairwise distances between sequences were exported from Geneious, then analyzed and visualized in RStudio (version 2022.2.0.443) (RStudio Team, 2022). Sequences with high intra-specific distances (>0.2) or low inter-specific distances (<0.02) were flagged as problematic and re-examined to determine potential causes.

Lastly, sequences from two fish families of interests, Lutjanidae (snappers), and Scombridae (tunas), were evaluated for clustering. These families were selected due to their significant economic and ecological roles in the WCPO (Hare et al., 2020; Newman et al., 2016; NOAA Fisheries, 2018), and both have been reported to present challenges in species discrimination using COI barcodes (Fadli et al., 2024; Hou et al., 2018; Victor, Valdez-Moreno & Vásquez-Yeomans, 2015). Sequences were aligned using MAFFT (Katoh, 2002), NJ trees were constructed with the Jukes-Cantor (JC) model (Jukes & Cantor, 1969), and pairwise distances were calculated in Geneious (version 2023.2.1). Distance data were visualized in low-dimensional space using t-Distributed Stochastic Neighbor Embedding (t-SNE) with the Rtsne package (version 0.17) (Krijthe, van der Maaten & Krijthe, 2018).

Results

Species checklist and barcode coverage

A total of 46,620 species records for the tropical region, 10,717 records for the north temperate region, and 40,892 records for the south temperate region, spanning nine marine phyla, were retrieved from OBIS for the WCPO. After data cleaning and dereplication, 26,682 records remained for the tropical region, 6,280 for the north temperate region, and 24,290 for the south temperate region, resulting in a combined total of 42,419 unique marine species records (Fig. 1A). The tropical region had the highest number of regionally distinct species (14,366 species, 33.9%), followed by the south temperate region (12,791 species, 30.2%). The north temperate region had the fewest distinct species (2,593 distinct species, 6.1%) (Fig. 1B).

Figure 1 Barcode coverage across nine marine phyla and geographic regions in the western and central Pacific Ocean (WCPO).

(A) Barcode coverage and species counts for nine marine phyla across north temperate, south temperate, and tropical regions of the WCPO. Red: species with no barcode records, green: species present in both BOLD and NCBI, yellow: species present only in BOLD, and blue: species present only in NCBI. The percentages indicate the proportion of species with barcode relative to the total number of species per phylum. (B) Proportion and barcode coverage for species categorized by distinct and shared distribution patterns among the north temperate, south temperate, and tropical regions of the WCPO. The inner donut plot illustrates the proportions of species in each distribution category, the outer donut plot illustrates the proportions of barcoded and non-barcoded species within each distribution category.

NCBI exhibited higher barcode coverage (31.5%, 13,346 species) than BOLD (29.8%, 12,621 species) (Table 1). Although the difference was statistically significant, the absolute difference in coverage was relatively small (1.7%). Among barcoded species, 78.2% had records in both databases, 13.4% had records only in NCBI, and 8.4% had records only in BOLD. Barcode coverage varied significantly across the north temperate, south temperate, and tropical regions (two-way ANOVA: F(2,103) = 7.3e+29, p < 0.001,) (Fig. 1A). The north temperate region had the highest barcode coverage (61.1%), followed by the tropical (41.0%) and south temperate (35.5%) regions. Marine phyla also exhibited significant differences in barcode coverage (two-way ANOVA: F(8,97) = 1.6e+30, p < 0.001) (Fig. 1A). Chordata had the highest barcode coverage (64.1%), followed by Echinodermata (41.4%), and Cnidaria (30.7%). In contrast, Mollusca (27.8%), Annelida (26.5%), and Arthropoda (26.0%) had similar barcode coverage levels, while Bryozoa (17.9%), Platyhelminthes (14.6%), and Porifera (14.4%) showed the lowest barcode coverage. Species occurring across all three regions had the highest barcode coverage (83.5%). In contrast, species distinct to a single region had the lowest barcode coverages, and species distinct to the south temperate region exhibited the lowest barcode coverage (17.9%) (Fig. 1B). Detailed species checklist and barcode coverage is available in File S1.

Table 1 Barcode coverage and quality comparisons between NCBI and BOLD.

	NCBI	BOLD	Two-proportions Z-test comparison between NCBI and BOLD	
			df	z-score	χ 2	p-value	
Barcode coverage	31.5%	29.8%	1	5.3	29.0	<0.01	
Over-represented species	3.7%	2.6%	1	22.7	517.4	<0.01	
Under-represented species	40.7%	38.8%	1	14.2	202.0	<0.01	
Long sequences (>1,000 bp)	2.8%	8.7%	1	97.4	9,480.3	<0.01	
Short sequences (<200 bp)	0.2%	 0%	1	21.1	444.4	<0.01	
Sequences with ambiguous nucleotides	3.1%	3.9%	1	16.1	258.8	<0.01	
Records missing taxonomic information	7.0%	3.4%	1	57.8	3,344	<0.01	
Conflicting records	0.27%	0.31%	1	3.0	9.14	0.02	
High intra-specific distances (>0.2)	36.7%	6.4%	1	30.0	902.8	<0.01	
Low inter-specific distances (<0.02)	2.9%	 0%	1	9.5	90.5	<0.01	

Barcode quality evaluation

A total of 321,997 sequences were retrieved from NCBI and 229,943 sequences from BOLD for WCPO species. NCBI contained more barcode sequences than BOLD for all marine phyla except Echinodermata (Figs. 2A, 2B).

Figure 2 Evaluation on NCBI (left) and BOLD (right) barcode reference databases for WCPO species.

(A–B) Number of sequences per phylum; (C–D) Proportion of species in each representation category; (E–F) Distribution of sequence length; (G–H) Proportion of sequences with ambiguous nucleotide characters; (I–J) Proportion of sequence records without taxonomic information at each taxonomic rank.

NCBI had a significantly higher proportion of over-represented and under-represented species (Fig. 2C), short sequences (Fig. 2E), and records with missing taxonomic information (Fig. 2I) compared to BOLD (Figs. 2D, 2F, 2J; Table 1). In contrast, BOLD had a higher proportion of long sequences (Fig. 2F) and sequences with ambiguous nucleotides (Fig. 2H) compared to NCBI (Figs. 2E, 2G; Table 1). However, NCBI had more sequences with ambiguous nucleotides located in the middle of the sequence (9,174) than BOLD (5,926). Notably, BOLD had no sequences shorter than 200 bp.

Among the phyla, Bryozoa had the highest proportion of under-represented species in both databases (67.9% in NCBI and 77.9% in BOLD), while Chordata had the lowest (26.3% in NCBI and 30.4% in BOLD) (Figs. 2C, 2D). Chordata had the highest number of short or long sequences (4,097 sequences) in NCBI (Fig. 2E) while Mollusca accounted for the largest number of long sequences (10,841 sequences) in BOLD (Fig. 2F). Porifera exhibited the highest proportion of sequences with ambiguous characters (11.7% in NCBI and 10.5% in BOLD), followed by Bryozoa (7.4% in NCBI and 6.9% in BOLD) (Figs. 2G, 2H). Mollusca had the highest proportion of records with missing order names in both databases (12.7% in NCBI and 11.5% in BOLD) (Figs. 2I, 2J).

Barcode taxonomic validation

The proportions of conflicting records were comparable between NCBI and BOLD (Fig. 3; Table 1). Both of NCBI and BOLD exhibited the highest number of conflicts at the species level. Upon further examination, conflicts caused by invalid names or synonyms accounted for 16.0% in NCBI and 22.5% in BOLD. Only five of these records (less than 1%) were submitted before 2010, suggesting that record age is unlikely to be a major contributor to the observed quality issues in this dataset. Among phyla, Chordata had the highest proportion of conflict records in both databases (29.4% in NCBI and 37.2% in BOLD). Platyhelminthes had the fewest conflict records, with only one instance in NCBI and none in BOLD. Detailed information about conflict records is provided in the File S2.

Figure 3 Number of sequence records with conflict taxonomic information, colored by phylum.

The y-axis indicates number of conflict records, the x-axis indicates taxonomic rank levels. Top, number of conflict records in NCBI; Bottom, number of conflict records in BOLD.

A total of 4,125 sequences from 100 species in NCBI and 3,119 sequences from the same 100 species in BOLD were selected for barcoding gap analysis. After calculating genetic distances, 19 outliers in NCBI and 20 outliers in BOLD exhibited extremely high genetic distances (≥ 2,000) with other sequences. After alignment and visual examination, these sequences were out of the typical barcode region, thus the outliers were excluded from further analysis.

NCBI sequences exhibited an average intra-specific distance of 0.10 ± 0.29 and an inter-specific distance of 0.46 ± 0.41 (Fig. 4A). BOLD sequences showed significantly lower average intra-specific distance (0.01 ± 0.04) and similar inter-specific distance (0.43 ± 0.11) (Fig. 4B). NCBI had significantly higher proportion of sequences with high intra-specific distances (i.e., sequences with family-level or higher taxonomic differences were labeled as the same species) and low inter-specific distances (i.e., sequences with lower than species-level differences were in different taxa) than BOLD (Table 1). Notably, BOLD had no case of low inter-specific distances.

Figure 4 Distribution of inter-specific distances and intra-specific distances in 100 species subsets of NCBI (left panel) and BOLD (right panel).

The y-axis indicates pairwise genetic distance between sequences, x-axis indicates frequency of distances, colored by inter-specific distance (red) and intra-specific distance (blue). (A–B) overall distance distribution in NCBI and BOLD; (C–D) phylum-focused distance distribution.

Among phyla, Bryozoa exhibited the highest proportion of sequences with large intra-specific distances (95.7% in NCBI and 96.0% in BOLD), followed by Cnidaria (56.8% in NCBI and 4% in BOLD) (Figs. 4C, 4D). At species level, Bugula neritina (Bryozoa) was most frequently associated with high intra-specific distance problems (File S2). Majority of sequences with low inter-specific distance problems showed conflicts at the phylum level (91.4%). Upon visual examination, these problematic sequences were likely to be false annotated or contamination sequences generated in the same project. Notably, 92.5% of BOLD records with high intra-specific distances were assigned to multiple BINs, while the remaining 7.5%, all from Bugula neritina, were still grouped within a single BIN. Detailed information about high intra-specific distances and low inter-specific distances records is provided in the File S2.

A total of 1,520 sequences from Lutjanidae and 1,962 sequences from Scombridae in NCBI, as well as 712 sequences from Lutjanidae and 1,190 sequences from Scombridae in BOLD were used for clustering analysis. The mean sequence distance for Lutjanidae was 0.18 ± 0.09 in NCBI (Fig. 5A) and 0.16 ± 0.09 in BOLD (Fig. 5B), and the mean distance for Scombridae was 0.15 ± 0.13 in both databases (Figs. 5C, 5D). Clustering plots for both databases revealed a consistent scatter pattern for Lutjanidae (Figs. 5A, 5B), indicating issues with high intra-specific distances. In contrast, Scombridae exhibited mixing pattern among species, suggesting low inter-specific distances in both NCBI and BOLD (Figs. 5C, 5D).

Figure 5 T-SNE clustering of Lutjanidae (A–B) and Scombridae (C–D) sequences in NCBI (left panel) and BOLD (right panel), colored by species.

Distribution of inter-specific distances and intra-specific distances for each group is showed on top-left of each figure. Detailed legend available in File S3.

Discussion

Barcode coverage and quality in NCBI and BOLD

In this study, NCBI demonstrated higher COI barcode coverage than BOLD, consistent with previous findings (Ardura, 2019; Duarte, Vieira & Costa, 2020; Hestetun et al., 2020). This difference can be attributed to NCBI’s status as a global-scale and public-accessible database, which allows users worldwide to freely upload sequences (Gold et al., 2021; Leray et al., 2019; Turanov & Kartavtsev, 2021). However, the absolute difference in barcode coverage between NCBI and BOLD was only 1.7%, which is unlikely to substantially impact biodiversity assessments, especially given that BOLD was observed to have fewer quality issues than NCBI (Table 1, Fig. 2). A more critical concern identified in this study is that more than two-thirds of marine species in the WCPO lack reference COI barcode records in both databases. This substantial gap currently limits the reliability of DNA barcoding for biodiversity monitoring in the WCPO, suggesting that further efforts to expand and curate reference databases are critical prerequisites for effective large-scale applications.

Our evaluation revealed key differences between BOLD and NCBI that users should consider when conducting metabarcoding studies. First, a larger proportion of WCPO species were either over-represented or under-represented in NCBI compared to BOLD. Second, NCBI contained a greater number of extremely short sequences (<200 bp), which may compromise barcode alignment accuracy. Third, missing taxonomic information was more prevalent in the NCBI records. Finally, while NCBI had a lower overall proportion of sequences with ambiguous nucleotide characters, a higher percentage of these problematic sequences had ambiguities in the central region of sequences, raising concerns about sequence quality. These issues likely stem from the original purposes of the two databases: NCBI was initially constructed for biomedical research (Benson, Lipman & Ostell, 1993), whereas BOLD was designed specifically for biodiversity applications (Ratnasingham & Hebert, 2007). A key contributor to BOLD’s higher reliability is its systematic use of voucher specimens, which are often lacking in NCBI records. Voucher-linked records not only enhance taxonomic accuracy but also facilitate curation and quality control, thereby improving the utility of BOLD for biomonitoring purposes (Heller et al., 2018; Ip et al., 2019; Valdez-Moreno et al., 2019). The utility of BOLD for large-scale biodiversity biomonitoring has been demonstrated by Valdez-Moreno et al. (2019), who employed thousands of voucher-linked BOLD records for eDNA-based assessments of fish community composition in a tropical oligotrophic lake.

The barcode evaluation workflow developed in this study highlighted several critical aspects affecting database reliability. For instance, species representation analysis identified disparities in barcoding and sequencing efforts with over-represented species indicating potential redundancy and under-represented species lacking sufficient sequence variation for reliable taxonomic assignment (Ardura, 2019; Bazinet et al., 2018; Costa et al., 2012; Weigand et al., 2019). Although NCBI exhibited broader barcode coverage overall, species representation was highly uneven, with some taxa disproportionately well-represented while others remained poorly represented. Targeted efforts to improve barcode coverage for underrepresented groups are essential to enhance the accuracy and completeness of both NCBI and BOLD databases (Ardura, 2019; Bazinet et al., 2018; Ramirez et al., 2020).

Sequence length evaluation is a crucial initial step in identifying problematic sequences that fall outside the target barcode region. Short sequences may result from sequencing artifacts and lead to ambiguous alignments (Nagai et al., 2022; Preston, Fritzsche & Woodcock, 2022), while excessively long sequences could contain non-target genomic regions or pseudogenes, which can interfere with barcode alignment (Guo et al., 2022; Song et al., 2008). BOLD, with a stricter quality control process (Ratnasingham & Hebert, 2007; Ratnasingham & Hebert, 2013), contained fewer short, low-quality sequences but a higher proportion of full-length COI gene sequences (1,500–1,600 bp) (Table 1, Fig. 2F). Although longer sequences may provide additional genetic information, trimming them to the barcode region is recommended to ensure consistency in analyses (Jeunen et al., 2023; Robeson et al., 2021).

The presence of ambiguous nucleotides, which can arise from sequencing errors, primer mismatches, or contamination is another critical issue (Redelings, 2014; Wheeler, 1994). Our analysis categorized ambiguities based on their positions (Figs. 2G–2H), which may provide insights into their potential causes and help guide appropriate handling strategies. For instance, ambiguities at sequence ends could be resolved by the users through trimming, while ambiguities in conserved regions require closer examination or exclusion from analyses depending on the purpose of the users. Although NCBI contained fewer ambiguous sequences overall, the higher proportion of centrally located ambiguities suggests potential sequencing quality issues compared to BOLD.

Missing taxonomic evaluations was another significant concern, reflecting potential issues such as outdated taxonomic classifications, unresolved species names, or user submission errors (Bouchet et al., 2017; Cunha & Giribet, 2019). NCBI exhibited more records lacking taxonomic hierarchy details, highlighting the need for improving curation and standardization. Addressing these deficiencies will require extensive database curation, ideally involving taxonomic specialists to ensure accurate species identification and consistent taxonomic frameworks across records.

Barcode taxonomic accuracy in NCBI and BOLD

Two major barcode taxonomic issues were identified in NCBI and BOLD: conflicting taxonomic records and high intra-specific genetic distances. Notably, all problematic records in BOLD lacked BIN numbers, reinforcing the BIN system’s utility in identifying and resolving errors.

Conflicting records, where identical or highly similar sequences were assigned to different taxonomic groups, were likely caused by (1) human errors or contamination during sequencing or data entry, particularly in datasets using broad-range primers designed for multiple taxa. Conflicts because of this reason were often observed between morphologically and/or ecologically distinct groups (Leray et al., 2019; Mugnai et al., 2021); (2) taxonomic inconsistencies, such as synonyms and invalid or outdated names (Fontes et al., 2021; Knebelsberger et al., 2014; Oliveira et al., 2016); (3) misidentifications, especially in morphologically similar taxa. For example, within Porifera, confusions between Axinellida and Bubarida was observed (File S2), likely due to their close morphologically resemblance and recent divergence history (Galitz et al., 2021).

The second major issue—high intra-specific distances—suggests the presence of cryptic species, sequencing contamination, or misidentifications. For example, sequences associated with the bryozoan Bugula neritina exhibited extreme genetic divergence, consistent with its known cryptic diversity and recent speciation events (Mackie, Keough & Christidis, 2006). The symbiotic relationships of Bugula neritina with other species (Linneman et al., 2014) may further contribute to the amplification of non-target DNA, potentially leading to ambiguous barcode assignments (Leray et al., 2019; Xie et al., 2025). These findings underscore the importance of involving taxonomic specialists to accurately resolve species boundaries. In addition, linking sequence records to voucher specimens would greatly support record validation and database curation efforts (Fontes et al., 2021; Ratnasingham & Hebert, 2007; Valdez-Moreno et al., 2019).

Sequence clustering plots provided a straightforward approach to visualize sequence distances, clusters, and outliers. Due to computational limitations, only two families were analyzed here. A more scattered pattern was observed in Lutjanidae, likely resulting from cross-phyla contamination, as some Lutjanidae sequences were identified as problematic in the barcoding gap analysis. For Scombridae, cross-species clustering highlighted inefficiencies in barcode resolution for distinguishing among Scombridae species, consistent with previous studies (Hou et al., 2018; Victor, Valdez-Moreno & Vásquez-Yeomans, 2015). These issues in Lutjanidae and Scombridae were consistent across both databases, underscoring the need for further optimization and improvement of DNA barcoding analyses for these taxa.

Barcode problems in the WCPO regions across nine marine phyla

The south temperate WCPO exhibited the lowest barcode coverage among the three regions, with over 80% of distinct species remaining unbarcoded. Severe barcode deficiencies were also noted in the tropical WCPO, particularly among distinct species. Given the high biodiversity in these regions, such barcode deficiencies could have serious consequences, including reduced ability to identify species, discover new or cryptic species, monitor biodiversity, or construct accurate phylogenetic profiles (Bucklin, Steinke & Blanco-Bercial, 2011). Notably, many countries in the Southern Hemisphere are economically disadvantaged (Sachs, Mellinger & Gallup, 2001), which limits their access to high-quality biodiversity investigation technologies. Collaborative initiatives between high-income and lower-income countries could play a critical role in expanding regional barcoding capacity and improving the representation of biodiversity hotspots within reference databases.

Additionally, the lack of barcode records for distinct species might suggest that sequencing resources and efforts may be disproportionately allocated to common and widely distributed species. Addressing this imbalance requires a critical increase in sequencing efforts targeting distinct and rare species. These species are essential for comprehensive biodiversity analyses and may play vital roles in maintaining ecosystem stability and functionality (Burlakova et al., 2011; Lamoreux et al., 2006).

Among phyla, Porifera, Bryozoa, and Platyhelminthes exhibited both low species counts and barcode coverage, requiring urgent attention to species inventories and molecular sequencing. Barcode quality issues were also notable in Porifera and Bryozoa, including a high proportion of sequences with ambiguous nucleotide characters and a lack of distinct COI barcoding gaps. These challenges may stem from high divergence rates (Linneman et al., 2014), cryptic diversity (Bucklin, Steinke & Blanco-Bercial, 2011; Mackie, Keough & Christidis, 2006), and symbiotic associations with other organisms (Linneman et al., 2014; Mugnai et al., 2021; Vargas et al., 2012), which might reduce barcode efficiency and reliability.

Despite the relatively high barcode coverage in Chordata, quality issues persisted, such as unreliable short sequences and conflicting records. Fish species, in particular, are often studied using broad-range primers (Iwasaki et al., 2013; Leray et al., 2013; Miya et al., 2015), increasing the risk of co-amplification and contamination. Additionally, some studies suggest that COI lacks sufficient species-level resolution for certain Scombridae species (Leray et al., 2019; Victor, Valdez-Moreno & Vásquez-Yeomans, 2015; Wangensteen et al., 2018), as observed in our clustering study. A significant issue in Mollusca was missing taxonomic information, likely due to taxonomic uncertainties and validations in certain Gastropoda groups (Bouchet et al., 2017; Cunha & Giribet, 2019). The absence of taxonomic information underscores the need for further database curation, phylogenetic analyses, and the consistent use of updated taxonomic names and classifications for molluscs, ideally involving collaboration with taxonomic experts and the use of authoritative resources such as WoRMS or MolluscaBase.

Our study is among the first to systematically evaluate both the coverage and quality of COI barcode records for marine metazoan species in the WCPO. Similar issues with reference databases have been reported in other regions (Mugnai et al., 2021; Ramirez et al., 2020), suggesting that these challenges are not region-specific. Therefore, our findings have broader implications beyond the WCPO, highlighting the need for global improvements in barcode database curation, quality control, and standardization to enhance the reliability of DNA barcoding for biodiversity research and conservation.

Recommendations for database curation

To improve reference database reliability, we propose the following curation strategies for database users:

1. Remove or trim sequences that fall outside the standard COI barcode length.

2. Discard sequences with ambiguous nucleotides in conserved regions and trim ambiguous sites and sequence ends.

3. Standardize taxonomic metadata by replacing missing taxonomic information with meaningful strings or marks (e.g., uncertain order name), and fill gaps by checking authorized species list to ensure taxonomic completeness.

4. Dereplicate sequences to minimize the effects of over-represented species and redundant sequences.

5. Flag and remove sequences with conflicting taxonomy, particularly those with discrepancies at higher taxonomic levels. When relevant, also consider the year of the record, as older sequences may reflect outdated taxonomic concepts or low record quality.

6. Conduct barcode gap analyses and clustering assessments to identify problematic sequences and assess barcode resolution.

Conclusions

Reliable reference databases are essential for the accuracy of DNA barcoding studies. However, a significant knowledge gap remains in the systematic evaluation of these databases, particularly in assessing both barcode coverage and quality of COI barcode sequences in NCBI and BOLD, focusing on marine metazoan species in the WCPO region.

Our findings indicate that NCBI exhibits higher barcode coverage than BOLD, which may be attributed to its broader research scope and less stringent submission requirements. In contrast, although BOLD is also open-access, its stricter quality control protocols and focus on biodiversity-specific data contribute to its relatively lower barcode coverage. However, this higher coverage comes at the cost of lower sequence quality, including a greater prevalence of short sequences, ambiguous nucleotides, and incomplete taxonomic information. In constrast, BOLD maintains a stricter quality control, but its reliance on curated metadata limits the number of submitted sequences, resulting in higher barcode deficiencies. The BIN system in BOLD demonstrates significant potential for identifying and addressing these problematic records, highlighting the advantages of curated databases.

Despite these differences, both databases contain a substantial proportion of problematic sequences, likely caused by contamination, cryptic species, sequencing errors, or inconsistent taxonomic assignments. Additionally, we identified severe barcode deficiencies in underrepresented marine regions—particularly in the south temperate WCPO—and among certain phyla such as the Bryozoa, Porifera, and Platyhelminthes. Even though the COI gene is widely regarded as a standard barcode for metazoan species, challenges remain in achieving clear species-level resolution for certain groups like Scombridae and Lutjanidae.

By addressing the critical gaps in barcode coverage, taxonomic representation, and sequence quality identified in this study, future barcoding initiatives can significantly enhance biodiversity monitoring and conservation research in the WCPO and beyond. This study underscores the importance of standardized protocols for database curation and sequencing practices, offering a roadmap for improving the reliability and applicability of DNA barcoding globally.

Supplemental Information

Supplemental Information 1 WCPO species checklist and barcode coverage

Supplemental Information 2 List of records with conflicts

Supplemental Information 3 Supplemental Figures

We thank Xing Quan for his assistance in analyzing data and creating figures.

Additional Information and Declarations

Competing Interests

Author Contributions

Data Availability

The authors declare there are no competing interests.

Yufei Zhou conceived and designed the experiments, performed the experiments, analyzed the data, prepared figures and/or tables, authored or reviewed drafts of the article, and approved the final draft.

Alejandro Trujillo-González conceived and designed the experiments, authored or reviewed drafts of the article, and approved the final draft.

Simon Nicol conceived and designed the experiments, authored or reviewed drafts of the article, and approved the final draft.

Roger Huerlimann conceived and designed the experiments, authored or reviewed drafts of the article, and approved the final draft.

Stephen D. Sarre conceived and designed the experiments, authored or reviewed drafts of the article, and approved the final draft.

Dianne Gleeson conceived and designed the experiments, authored or reviewed drafts of the article, and approved the final draft.

The following information was supplied regarding data availability:

The script for the barcode evaluation workflow is available at GitHub and Zenodo:

- https://github.com/xyzzzeno/reference_evaluation.

- xyzzzeno. (2025). xyzzzeno/reference_evaluation: Code for Evaluation of DNA barcoding reference databases for marine species in the Western and Central Pacific Ocean (v1.0.0). Zenodo. https://doi.org/10.5281/zenodo.15644776

The raw barcode records are available in the Supplementary File.

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
