# Peer review of "Evaluation of DNA barcoding reference databases for marine species in the western and central Pacific Ocean"

_PeerJ, doi:10.7717/peerj.19674_

## Round 0.1 · original submission · Major Revisions

I agree with both reviewers that the topic important and the study’s overall design and statistical treatment are valid. However, they raise substantive concerns that must be addressed before the paper can be reconsidered for publication. Kindly address all concerns and suggestions raised.

Reviewer 1 ·

Basic reporting

This study is quite interesting and provides valuable annotations regarding two of the main databases used in biodiversity research. However, it appears that the authors lack a deep understanding of BOLD and its operational framework. Among the major concerns, I recommend that the authors focus exclusively on records with a BIN in BOLD, as these are the only ones that can be considered more or less reliable. All other records must undergo certain quality controls to be assigned a BIN. Although this approach would significantly reduce the number of BOLD records analyzed, it would greatly improve their quality. Any researcher familiar with BOLD would not consider comparing their data with records that lack a BIN. Another difference is that BOLD has a direct link to the voucher deposited in a scientific collection, and it is crucial because it will allow the curation of the database and a correction of all mistakes. In GenBank corrections are more difficult.
The statistical analyses employed are appropriate, and the queries are clearly explained, with all scripts uploaded to GitHub, which is highly appreciated.
Below are some specific comments regarding the manuscript:
• Lines 276–279: I suggest the authors concentrate solely on specimens with a BIN in BOLD. Each of these specimens has passed a series of validation steps to receive a BIN. I recommend a careful reading of the paper by Ratnasingham and Hebert (2013) for a deeper understanding. It is likely that all figures will change if the authors restrict their analyses to BIN-assigned records. Only sequences with a BIN should be considered "valid." This recommendation is reiterated for Lines 336–338.
• Lines 294–295: In my opinion, biomonitoring efforts should be postponed, as the databases are still in the process of being developed.
• Lines 306–310: Extensive curation of databases is still required, and this process necessitates the involvement of taxonomic specialists. This point is missing in the whole manuscript, and it will be essential for the quality of the data.
• Line 318: This statement is only partially accurate. BOLD can indeed host genes other than COI. Additionally, the authors have not considered that NCBI was originally created for medical purposes and later adapted for biodiversity data. In contrast, BOLD was specifically designed for biodiversity, which is a key distinction. Many older records in NCBI lack essential information lincked to the vouchers, required for biodiversity studies and its applications.
• Lines 333–334: This issue stems from NCBI’s adaptation to accommodate non-medical fields of study.
• Lines 348–353: The authors emphasize sequencing and databases but overlook the essential role of taxonomists. Most BOLD records should have a corresponding voucher specimen deposited in a scientific collection. In NCBI, such voucher-based validation is often lacking, which is crucial for verifying species identifications and develop the corresponding corrections in the databases.
• Lines 363–375: Unfortunately, many countries in the Global South are economically disadvantaged, which contributes to the deficient barcode coverage in these regions. The authors could suggest fostering collaborations between high-income and low- or middle-income nations to address this gap.
• Lines 384–386: The authors should consider the age of the sequences. Older sequences are more likely to present issues such as editing errors or problems related to primer design.
• Lines 423–424: BOLD is, in fact, an open-access database. It merely requires that contributors register, which is part of the platform’s quality assurance process. It is unclear why the authors claim that BOLD is not open. Anyone can contribute to BOLD, provided they register.
Lastly, for any specific taxonomic groups, BOLD allows for the creation of datasets containing thousands of specimens, as demonstrated in the following article published in PLOS ONE:
Valdez-Moreno, M., N. V. Ivanova, M. Elias-Gutierrez, S. L. Pedersen, K. Bessonov, and P. D. N. Hebert. 2019. "Using eDNA to biomonitor the fish community in a tropical oligotrophic lake." PLOS ONE 14(4). doi:10.1371/journal.pone.0215505.
This is a good example of a study that used eDNA for biomonitoring, leveraging one of the most accurate databases available at the time, involving thousands of specimens.
In conclusion, I believe the manuscript is promising, but the authors should revise it thoroughly to address the points and suggestions provided above before it can be considered for acceptance.

Experimental design

No comment, everything is included in the first part.

Validity of the findings

No comment

Additional comments

No comment.

·

Basic reporting

The manuscript is generally well-written and follows a logical structure, with clear figures and adequate referencing. However, some parts of the introduction and discussion sections require revision for clarity and scientific precision. Several sentences are awkwardly phrased (e.g., lines 299–300), and some claims are not properly supported by the references provided (e.g., lines 315–317 on sequence artifacts and pseudogenes). Certain references used do not match the claims made (e.g., Kennedy et al., 2020 for a general statement on metabarcoding), and more appropriate reviews or broader studies should be cited instead.

I also suggest tightening vague or redundant statements in the introduction (lines 50–52) and removing or revising parts that are speculative or unsupported by the data (e.g., line 351 on symbiosis and contamination).

Here are detailed notes:

Line 45 – Needs references to support the first sentence.
Line 45 and 46 – Modify to “(…) the use of a standardized short DNA fragment (…)” and it should be “biodiversity assessments” in plural.
Lines 48 to 50 – The phrase “Together with high-throughput sequencing technologies, DNA barcoding can be applied more efficiently at a community-level, which is known as DNA metabarcoding (Kennedy et al., 2020)." Should be improved for clarity. I suggest something along the lines of "When combined with high-throughput sequencing technologies, DNA barcoding can efficiently characterize entire biological communities, an approach known as DNA metabarcoding (Kennedy et al., 2020)."
Also regarding this reference, there are studies that better fit this sentence, that are not focused in a particular group (spiders - which is also not the aim of this study). I think one or two references for metabarcoding should be added, reviews even.
Lines 50-52: “Both DNA barcoding and metabarcoding are widely used in many ecological and biodiversity investigations (Clare, 2014; Günther et al., 2021; Leray et al., 2012; Shehzad et al., 2012). This statement doesn't inform the reader about anything specific, maybe you could add more information on the application of this technology with a few examples or just cut this sentence entirely.

Lines 59-60: “However, the accuracy and reliability of these records have been the subject of debate (Ardura, 2019; Leray et al., 2019; Shen et al., 2013; Turanov & Kartavtsev, 2021).” Confirm if these references mention other databases. From what I briefly read, these papers only study accuracy and reliability in NCBI. If possible, give at least one example per database.

Lines 74 – 76: "For instance, the Barcode Index Number (BIN) system (Ratnasingham & Hebert, 2013) is a convenient feature of BOLD to identify problematic records and enhances the reliability of sequence and taxonomy data (Costa et al., 2012; Fontes et al., 2021; Oliveira et al., 2016).” I think this sentence oversimplifies the BIN system, maybe you could give more context and mention how the BIN system clusters similar sequences that likely belong to the same species into a single BIN which facilitates species delimitation and it also allows the identification of problematic records (…). Just give a little more context on the BIN system to help the reader.

Line 78: “(…) taxonomic resolution and an increasing (…)” remove the “an”.

Line 82: These references need to be changed as these referenced papers do not show that BOLD has higher barcode deficiencies due to the reasons mentioned by the authors. Also, in lines 80-82 the authors should refer the comparison to other databases and justify it with the proper references.

Line 88: comprehensive, not comprehensively

Line 95: You should find a more accurate and region-specific reference, ideally one explicitly addressing the biodiversity and exploitation status of the WCPO. I don’t think this paper (at least alone) fits this sentence.

Lines 105-107: Feels more like results so I believe it should be eliminated from the introduction section.

Figure 1A: Just a suggestion to be easier to analyze the data, maybe you could put the numbers at the right of each bar, because in some bars the numbers don’t fit.

Figure 1B: Also a suggestion to put all the barcoded and non-barcoded legends on the outside instead of some on the outside and some on the inside.

Line 295: “species in the WCPO lack reference COI barcode records in both databases, This substantial gap” substitute the comma with a full stop.

Lines 299-230: “First, NCBI exhibited a higher proportion of WCPO species were either over- or under-represented to BOLD”. This sentence is poorly written, please rewrite to something in the lines of “First, a greater proportion of WCPO species were either over- or under-represented in NCBI compared to BOLD”.

Lines 311-313: “While NCBI displayed broader coverage, its records were more unevenly distributed across species. Enhancing barcode representation for underrepresented taxa is necessary to improve the effectiveness of both databases.” These sentences could use improving, the first is vague and should be more specific, while the second could be improved to something in the lines of “Targeted efforts to improve coverage for underrepresented groups are essential to enhance the accuracy and completeness of both NCBI and BOLD”. And maybe cite some papers that reach the same conclusions.

Lines 315-317: The references used for this sentence are not appropriate as neither paper discusses contaminations, sequence artifacts or pseudogenes. You should search for references that explain the phenomena you mention for shorted and longer barcode sequences.

Lines 318 – 320: “BOLD, as a COI-specific database with a stricter quality control process (Ratnasingham & Hebert, 2007, 2013), contained fewer short, low-quality sequences but a higher proportion of full-length COI gene sequences (1,500-1,600 bp).” Mention the figure or table or supplementary information from where you retrieved this information on your results.

Lines 320 – 322: You need to add references to this sentence.

Line 327: a comma is missing after “be resolved by the users through trimming”

Lines 339-344: You need to add more references in this paragraph to justify these possible reasons. There are reviews and papers on barcode data curation that explore these topics.

Lines 351-353: The sentence “The symbiotic relationships of Bugula neritina with other species (Linneman et al., 2014) further increase the risk of sequencing contamination.” Needs to be revised because having symbionts only increases the risk of sequence contamination in the case of use of bulk communities or eDNA. I also think that the term “sequencing contamination” needs to be more specific as this could refer to lab contamination, amplification of non-target DNA, etc. Please rewrite the sentence.

Lines 384-386: The sentence is duplicated, erase everything after “persisted” and erase the comma before “persisted”.

Lines 392-394: What do you mean by “standardization of taxonomic practices for molluscs”? Maybe be a little more specific in this portion of the sentence.

Experimental design

The methods are appropriate for the goals of the study and include a comprehensive barcode evaluation workflow. However, the rationale for certain methodological choices needs to be better explained. For example, the separation of WCPO regions (line 118) should be justified, and clarification is needed on whether the taxonomic validation was conducted using WoRMS in line 121.

Additionally, the selection of Lutjanidae and Scombridae for the clustering analysis should be more clearly justified, particularly in the context of WCPO species or relevance to barcoding issues.

Detailed notes:
Lines 118-120: Why did you choose to separate these regions of the WCPO?

Lines 121-123: Where these also checked on WoRMS?

Line 126: alter to “worms package” (remove the extra r)

Validity of the findings

The findings are generally valid and well-supported by the data, though some claims cannot be verified based on the current supplementary materials. For instance, Bugula neritina is stated to be the species most frequently associated with high intra-specific conflict, but I did not find it in Supplementary File 2. Additionally, the conflict data file does not distinguish between high intra- and low inter-specific distances, yet these are interpreted in the results (lines 272–277).

The statistical comparison between NCBI and BOLD barcode coverage (lines 211–212) shows a statistically significant but practically negligible difference (1.7%). The authors should clarify whether this difference is practically meaningful.

Furthermore, several biological interpretations (e.g., contamination due to symbionts, and causes of sequence length variation) are speculative or lack proper referencing and should be toned down or better supported. The sentence about ambiguity patterns (lines 324–326) refers to an analysis that is not supported by a figure or table and should either be documented or removed.

Detailed notes:

Lines 211-212: I understand that statistically the difference is significant, however the practical significance is small (only 1.7% difference), maybe you could clarify whether this difference is ecologically or practically meaningful, rather than just statistically significant. Because in practical terms the barcode coverage is only slightly higher…

Lines 253 – 256: I think the supplementary file2 should contain the data separately for each database as we cannot validate these percentages without that distinction.

Lines 272 – 273: “At species level, Bugula neritina (Bryozoa) was most frequently associated with high intra-specific distance problems (Supplementary file2).” Bugula neritina is not mentioned in this supplementary file. This supplementary file also does not include a column specifying whether the conflict is due to high intra-specific distance or low inter-specific distance, as such it does not act as a table to justify the paragraph from lines 273 to 277.

Lines 324-326: “Our analysis categorized ambiguities based on their positions, providing insights into their causes and guiding appropriate handling strategies.” There is no file or table on the document or supplementary materials that presents these results, I think this would be important to add to the manuscript.

Additional comments

This study addresses an important gap in DNA barcoding database quality, particularly within the WCPO region, where reference coverage remains limited. The authors develop a detailed barcode evaluation framework and apply it comprehensively to two major databases. Their findings and curation recommendations are timely and valuable for both database users and curators.

That said, the manuscript requires moderate revisions to clarify the methods, support key claims with appropriate data and references, and improve the quality and precision of the writing. I truly believe that with these improvements, the study will offer a strong and useful contribution to the field of molecular biodiversity monitoring, particularly for database curation protocols.

---

## Round 0.2 · accepted · Accept

Thank you, authors, for addressing all the concerns and comments raised by the reviewers.

Reviewer 1 ·

Basic reporting

The manuscript has been significantly improved; the authors adequately addressed all suggestions and enhanced the literature review.

I believe it can be accepted for publication in its current form, as it provides a thorough and well-structured assessment of the value and the need for improvement of the two main databases commonly used in studies involving standardized genes.

Experimental design

Although the study does not include an experimental design, the tools employed for the consultation and comparison of the two databases are appropriate and correctly applied.

Validity of the findings

This study provides a valuable assessment of the advantages, disadvantages, and reliability of the two main databases commonly used in genetic studies, focusing specifically on the Western and Central Pacific Ocean. Nonetheless, its findings can serve as a reference for other regions and taxonomic groups worldwide.

Additional comments

No comments

·

Basic reporting

After reviewing the revised manuscript I can see that the authors have made the changes suggested by the reviewers and I believe the manuscript is ready for publication.

Experimental design

After reviewing the revised manuscript I can see that the authors have made the changes suggested by the reviewers and I believe the manuscript is ready for publication.

Validity of the findings

After reviewing the revised manuscript I can see that the authors have made the changes suggested by the reviewers and I believe the manuscript is ready for publication.

Additional comments

After reviewing the revised manuscript I can see that the authors have made the changes suggested by the reviewers and I believe the manuscript is ready for publication.